

# Evaluation and identification of stem rust resistance genes *Sr2*, *Sr24*, *Sr25*, *Sr26*, *Sr31* and *Sr38* in wheat lines from Gansu Province in China

Xiao Feng Xu, Dan Dan Li[*], Yang Liu[*], Yue Gao, Zi Yuan Wang, Yu Chen Ma, Shuo Yang, Yuan Yin Cao, Yuan Hu Xuan and Tian Ya Li

College of Plant Protection, Shenyang Agricultural University, Shenyang, China
[*] These authors contributed equally to this work.

Corresponding authors
Tian Ya Li, litianya11@syau.edu.cn
Yuan Hu Xuan,
xuanyuanhu115@syau.edu.cn

## ABSTRACT

Wheat stem rust, caused by *Puccinia granimis* f. sp. *tritici*, severely affects wheat production, but it has been effectively controlled in China since the 1970s. However, the appearance and spread of wheat stem rust races Ug99 (TTKSK, virulence to *Sr31*), TKTTF (virulence to *SrTmp*) and TTTTF (virulence to the cultivars carrying *Sr9e* and *Sr13*) have received attention. It is important to clarify the effectiveness of resistance genes in a timely manner, especially for the purpose of using new resistance genes in wheat cultivars for durable-resistance. However, little is known about the stem rust resistance genes present in widely used wheat cultivars from Gansu. This study aimed to determine the resistance level at the seedling stage of the main wheat cultivars in Gansu Province. A secondary objective was to assess the prevalence of *Sr2*, *Sr24*, *Sr25*, *Sr26*, *Sr31*, and *Sr38* using molecular markers. The results of the present study indicated that 38 (50.7%) wheat varieties displayed resistance to all the tested races of *Puccinia graminis* f. sp. *tritici.* The molecular marker analysis showed that 13 out of 75 major wheat cultivars likely carried *Sr2*; 25 wheat cultivars likely carried *Sr31*; and nine wheat cultivars likely carried *Sr38*. No cultivar was found to have *Sr25* and *Sr26*, as expected. Surprisingly, no wheat cultivars carried *Sr24*. The wheat lines with known stem rust resistance genes could be used as donor parent for further breeding programs.

## INTRODUCTION

*Puccinia graminis* Pers. f. sp. *tritici* Eriks. and E. Henn (*Pgt*) causes one of the most potentially destructive wheat diseases, seriously threatening world grain production (*Pardey et al., 2013*). Disease-resistance breeding to control wheat stem rust is economic, effective, and protective of the environment, and has been proved to be the best control method by repeated practice (*Goutam et al., 2015*). Wheat stem rust has been effectively controlled with the wide use of resistance gene *Sr31* from a 1BL/1RS wheat–rye chromosome arm translocation (*Rouse et al., 2012*). However, a new race Ug99 virulent to *Sr31* was identified in Uganda and classifed as TTKS by the North American Nomenclature System of *Pgt* in 1999 (*Pretorius et al., 2000*). Ug99 has broad virulence, and mutates and spreads quickly.

Since 1999, 13 variants of Ug99 have been found in 13 countries (*FAO, 2017*). Recently, Ug99 has been monitored in Egypt, which is the main wheat production area of the Middle East, revealing that its mode of spread is similar to that of a virulent stripe rust pathogen race to *Yr9* predicted by Geographic Information System of CIMMYT (*Singh et al., 2006*). Following the identification and spread of the Ug99 race group, a new race TKTTF caused a wheat stem rust epidemic with an estimated 20,000 to 40,000 ha likely planted to 'Digalu' (with resistance to Ug99 race group) in Southern Ethiopia during 2013–2014 (*Olivera et al., 2015*). Currently it has been confirmed in 11 countries, and given the rapid and destructive nature of race TKTTF, close monitoring of this race is advised—especially in countries which have cultivars carrying the *SrTmp* resistance gene.

A new race TTTTF with virulence to *Sr9e* and *Sr13* attacked thousands of hectares of durum wheat in Sicily, Italy in 2016, resulting in the largest burst of wheat stem rust in Europe since the 1950s (*Bhattacharya, 2017*). The large number of spores produced by TTTTF may continue the epidemic in 2017. Moreover, the researchers from the Global Rust Research Center shared a major concern in the warning report that TTTTF could infect not only durum wheat and bread wheat but also dozens of laboratory-grown strains of wheat (*FAO, 2017*). In view of this, in February 2017, *Nature* highlighted the potential threat to European wheat production of this race (*Bhattacharya, 2017*). Therefore, the spread of Ug99, TKTTF and TTTTF, and their variants, threaten the wheat production safety in China.

Gansu Province, located in the northwest of China, plays a significant role in the epidemic and spread of wheat stem rust in China (*Cao, 1994*). Resistance breeding for this disease has not been a primary objective because it has been effectively controlled in China since the 1970s (*Wu et al., 2014*). However, durable resistance to stem rust has been re-emphasized with the occurrence and spread of new races of *Pgt*. It is necessary to analyze the resistance genes in wheat cultivars (lines) from Gansu Province, and the information provided here will be important for developing potentially durable combinations of stem rust resistance genes in cultivars.

## MATERIALS AND METHODS

### Wheat cultivars and near-isogenic lines

A total of 75 tested wheat cultivars in Gansu Province were provided by Dr. Fangping Yang from the Wheat Research Institute, Gansu Academy of Agricultural Sciences.

Molecular markers linked to six *Sr* genes were tested: *Sr2*, *Sr24*, *Sr25*, *Sr26*, *Sr31*, and *Sr38*. Near-isogenic lines carrying 45 *Sr* genes were used to confirm the validity of these molecular markers. The near-isogenic lines carrying these resistance genes were provided by Dr. Yue Jin from USDA-ARS, Cereal Disease Laboratory, University of Minnesota, USA.

The tested *Pgt* races included the 21C3CTHTM, 21C3CFHQC, 34MKGQM, 34MKGSM, 34C3MTGQM and 34C3RTGQM (race 34C3MTGQM and 34C3RTGQM identified from the alternative host, *Berberis*). These races were named according to the methods described in a published study (*Li et al., 2016b*). The full names of the races and their virulence/avirulence patterns are shown in Table 1. They were isolated and identified by the Plant Immunity Institute, Shenyang Agricultural University, China.

**Table 1** Virulence/avirulence patterns of six races of *P. graminis* f. sp. *tritici*.

| Race | Ineffective *Sr* genes | Effective *Sr* genes |
|---|---|---|
| 21C3CTHTM | *6, 7b, 8a, 9a, 9b, 9d, 9f, 9g, 10, 11, 12, 13, 14, 15, 16, 17, 18, 24, 28, 29, 34, 35, Tmp, McN* | *5, 9e, 19, 20, 21, 22, 23, 25, 26, 27, 30, 31, 32, 33, 36, 37, 38, 47* |
| 21C3CFHQC | *7b, 8a, 9a, 9b, 9d, 9f, 9g, 12, 13, 14, 15, 16, 17, 18, 28, 29, 34, 35, McN* | *5, 6, 9e, 10, 11, 19, 20, 21, 22, 23, 24, 25, 26, 27, 30, 31, 32, 33, 36, 37, 38, 47, Tmp* |
| 34MKGQM | *5, 6, 7b, 8a, 9a, 9b, 9d, 9f, 9g, 12, 15, 16, 20, 24, 27, 28, 29, McN* | *9e, 10, 11, 13, 14, 17, 18, 19, 21, 22, 23, 25, 26, 30, 31, 32, 33, 34, 35, 36, 37, 38, 47, Tmp* |
| 34MKGSM | *5, 6, 7b, 8a, 9a, 9b, 9d, 9f, 9g, 10, 12, 15, 16, 20, 24, 27, 28, 29, McN* | *9e, 11, 13, 14, 17, 18, 19, 21, 22, 23, 25, 26, 30, 31, 32, 33, 34, 35, 36, 37, 38, 47, Tmp* |
| 34C3RKGQM | *5, 6, 7b, 8a, 9a, 9b, 9d, 9f, 9g, 12, 16, 19, 21, 23, 24, 27, 28, 29, McN* | *9e, 10, 11, 13, 14, 15, 17, 18, 20, 22, 25, 26, 30, 31, 32, 33, 34, 35, 36, 37, 38, 47, Tmp* |
| 34C3MTGQM | *7b, 8a, 9a, 9b, 9d, 9f, 9g, 11, 12, 13, 14, 15, 16, 17, 18, 28, 29, 34, 35, McN* | *5, 6, 9e, 10, 19, 20, 21, 22, 23, 24, 25, 26, 27, 30, 31, 32, 33, 36, 37, 38, 47, Tmp* |

## Seedling resistance evaluation

The cultivars were planted in porcelain pots with a 12-cm-diameter. Seven days later, the leaves were moistened by water with 0.1% Tween 20 using an atomizer and then sprayed with 1 g of fresh urediniospores and dried talc in a ratio of 1:20 (v:v). The inoculated seedlings were transferred to a greenhouse with the temperature in a range of 18 to $22 \pm 1\,°C$. Three biological replicates of the seedling assays were performed for each *Pgt* race. After 14 days of inoculation, the infection types (ITs) were recorded using the 0–4 IT scale (*Stakman, Stewart & Loegering, 1962*). ITs were then grouped into low ('0', ';', '1', '1+', '2′', '2+', and X) and high ('3–', '3', '3+', and '4') infection types. The ITs used in this study are shown in Fig. 1.

## DNA extraction

DNA was extracted from young leaves of 10-day-old seedlings using a genomic DNA extraction kit (http://www.sangon.com/, China). The DNA quality was examined by 1.2% (w/v) agarose gels and DNA quantification was performed using the NanoDrop-1000 version 3.3.1 spectrophotometer.

Polymerase chain reaction (PCR)-specific primers were synthesized by Shanghai Biotech Biotech Co., Ltd, China (Table 2). PCR amplifications were carried out in 25 µL volume, including 0.5 µL of 10 mmol $L^{-1}$ deoxyribonucleoside triphosphates, 2.5 µL of 10× buffer ($Mg^{2+}$), 0.2 µL of 5 U µ$L^{-1}$ Taq polymerase, 1 µL of 10 µmol $L^{-1}$ of each primer, and 2 µL of 30 ng µ$L^{-1}$ DNA. De-ionized water was used to achieve 25 µL volume. Condition of PCR amplification were as follows: 94 °C for 4 min, 30 cycles of 94 °C for 45 s, 60 °C for 45 s, and 72 °C for 1 min, followed by the final extension at 72 °C for 8 min; other specific conditions were as described in previous studies (Table 1).

# RESULTS

## Wheat seedling resistance

The resistance test results of 75 main wheat cultivars in Gansu to the races 21C3CTHTM, 21C3CFHQC, 34MKGQM, 34MKGSM, 34C3MTGQM, and 34C3RTGQM are shown in

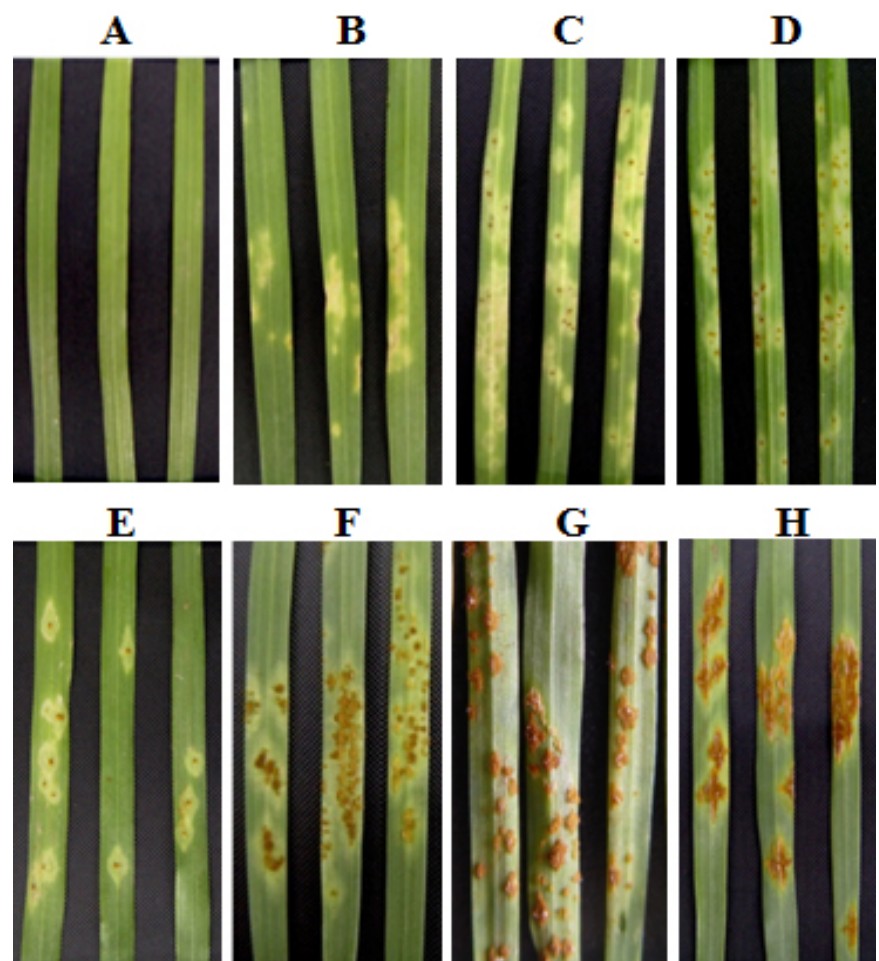

**Figure 1 Infection types (ITs) used in this study.** A–H indicate ITs: 0, ;, ;1, 1, 2, 3-, 3, 4.

**Table 2 The markers linked to resistance genes *Sr2, Sr24, Sr26, Sr31* and *Sr38* with their forward and backward primers.**

| Genes | Marker | Forward primer | Reverse primer | References |
|---|---|---|---|---|
| Sr2 | *Xgwm533* | 5′-GTTGCTTTAGGGGAAAAGCC | 5′-AAGGCGAATCAAACGGAATA | *Hayden, Kuchel & Chalmers (2004)* |
| | *csSr2* | 5′-CAAGGGTTGCTAGGATTGGAAAAC | 5′-AGATAACTCTTATGATCTTACATTTTTCTG | *Mago et al. (2011)* |
| Sr24 | *Sr24#12* | 5′-CACCCGTGACATGCTCGTA | 5′-AACAGGAAATGAGCAACGATGT | *Mago et al. (2005)* |
| | *Sr24#50* | 5′-CCCAGCATCGGTGAAAGAA | 5′-ATGCGGAGCCTTCACATTTT | *Mago et al. (2005)* |
| Sr25 | *Gb* | 5′-CATCCTTGGGGACCTC | 5′-CCAGCTCGCATACATCCA | *Liu et al. (2010)* |
| Sr26 | *Sr26#43* | 5′-AATCGTCCACATTGGCTTCT | 5′-CGCAACAAAATCATGCACTA | *Mago et al. (2005)* |
| Sr31 | *SCSS30.2$_{576}$* | 5′-GTCCGACAATACGAACGATT | 5′-CCGACAATACGAACGCCTTG | *Das et al. (2006)* |
| | *Iag95* | 5′-CTCTGTGGATAGTTACTTGATCGA | 5′-CCTAGAACATGCATGGCTGTTACA | *Mago et al. (2002)* |
| Sr38 | *VENTRIUP-LN2* | 5′-AGGGGCTACTGACCAAGGCT | 5′-TGCAGCTACAGCAGTATGTACACAAAA | *Helguera et al. (2003)* |
| | *URIC-LN2* | 5′-GGTCGCCCTGGCTTGCACCT | 5′-TGCAGCTACAGCAGTATGTACACAAAA | *Helguera et al. (2003)* |

**Table 3  Resistant proportion of 75 wheat cultivars to six races of *P. graminis* f. sp. *tritici*.**

| Races | Susceptible | | Resistance | |
|---|---|---|---|---|
| | Number of cultivars | Percentage/% | Number of cultivars | Percentage/% |
| 21C3CTHTM | 28 | 37.3 | 47 | 62.7 |
| 21C3CFHQC | 25 | 33.3 | 50 | 66.7 |
| 34MKGQM | 30 | 40.0 | 45 | 60.0 |
| 34MKGSM | 26 | 34.7 | 49 | 65.3 |
| 34C3RKGQM | 26 | 34.7 | 49 | 65.3 |
| 34C3MTGQM | 25 | 33.3 | 50 | 66.7 |
| All tested races | 37 | 49.3 | 38 | 50.7 |

Table 3. Thirty-eight (50.7%) of the 75 tested wheat cultivars showed different resistance levels (ITs 0, ;, ;1, 1+, and 2) to the six races at the seedling stage (Table 4). The remaining 38 (50.7%) wheat cultivars showed varying levels of susceptibility (ITs 3, 3−, 3+, and 4) (Table 3).

## Validity of the markers

Six specific PCR markers closely linked with resistance genes *Sr2*, *Sr24*, *Sr25*, *Sr26*, *Sr31*, and *Sr38* were validated using 45 single differentials carrying known resistance genes. Table 5 shows that these ten markers amplified only specific bands in the expected wheat genetic stocks. For example, primer $SCSS30.2_{576}$ amplified only 576-bp specific bands in Siouxland, Sisson, Sr31/6*LMPG, and Federation*4/Kavl, while in other wheat lines without *Sr31*, no bands were amplified, indicating that these markers are able to be well applied for the molecular detection of the six resistance genes.

### *Sr2* screening

A DNA marker was developed to accurately predict *Sr2* in diverse wheat germplasm for the partial resistance of *Sr2* is very difficult to screen under field conditions (*Mago et al., 2011*). Two markers, *Xgwm533* and *csSr2*, were used to detect *Sr2* in wheat cultivars of Gansu Province. A specific PCR band with 120-bp in size was amplified with marker *Xgwm533*, but no PCR product was amplified using marker *csSr2* in Hope with *Sr2*. In this study, a similar 120-bp band was detected in the 13 cultivars, indicating that these cultivars carried *Sr2* (Table 6).

### *Sr24* screening

Two markers, *Sr24#12* and *Sr24#50*, were developed to detect *Sr24*, located on chromosome 3DL in Agent- or 1BS in Amigo-derived lines (*Mago et al., 2005*). These two markers were applied to detect *Sr24* existence in the 75 major wheat cultivars (lines) of Gansu Province in this study. The results showed that marker *Sr24#12* amplified a 500-bp specific band and marker *Sr24#50* amplified an approximately 200-bp specific band in the *Sr24* control Lc*Sr24*Ag. No PCR fragment was amplified in Little Club (LC) and the tested cultivars, indicating that these cultivars lacked *Sr24*.

Xu et al. (2017), PeerJ, DOI 10.7717/peerj.4146

**Table 4  Seedling infection types produced by six races of *P. graminis* f. sp. *tritici* on 75 wheat cultivars (lines).**

| Cultivars (lines) | Pedigree | Infection types[a] | | | | | |
|---|---|---|---|---|---|---|---|
| | | 21C3CTHQM | 21C3CFHQC | 34MKGQM | 34MKGSM | 34C3RTGQM | 34C3MTGQM |
| Ningchun 39 | Yong 833/Ningchu 4 | 0 | 1 | 0 | 1 | 0 | 0 |
| Dingfeng 10 | Tal 73-3/Mota | 0 | 0 | 0 | 1 | 0 | ; |
| Linmai 32 | Ganfu 92-310/Xianyang-dasui | 4 | 4 | 4 | 3− | 3 | 4 |
| Wuchun 8 | Shi 1269/Shi 1269 | 1+ | 0 | 3− | 1 | 0 | 0 |
| Wuchun 7 | Yong 434/Jian 94-114 | 4 | 3− | 4 | 1 | 4 | 3− |
| Dingxi 41 | 8124-10/Dongxiang 77-011 | ; | 0 | 0 | ; | 0 | 0 |
| Longchun 31 | Genic male sterility of Taigu | 0 | ;1 | 0 | 1 | 0 | ; |
| Longchun 22 | CHIL/BUC | 0 | 0 | 0 | 3 | 0 | 2 |
| Ganchun 25 | M34IBWSN-262/M34IBWSN-252// Zhangchun 11/Yongliang 4 | 0 | 0 | 0 | 0 | 0 | 0 |
| Longchun 25 | Yong 1265/Corydon | ; | 1 | 2 | 0 | 2 | 0 |
| Longchun 23 | Introduced from CIMMYT | 0 | 1 | 0 | 1+ | 0 | ; |
| Longchun 26 | Yong 3263/Gaoyuan 448 | 0 | 0 | 0 | 1 | 0 | ; |
| Ganchun 24 | Zhangchun11/93-7-31//23416-8-1//Aibai/ Kavkaz | 0 | 2 | 0 | ; | 0 | ; |
| Yinchun 9 | Dingxi 35/Xihan 1//Dingxi 37/9208 | 0 | 0 | 0 | 0 | 0 | 2 |
| Longchun 28 | 8858-2/Longchun 8 | ; | 1 | 0 | 3 | ; | 3 |
| Wuchun 5 | 7906/ROBLIN//21-27 | 1+ | 1 | 3 | 4 | ; | 3− |
| Ganchun 20 | 88-862/630 | 4 | 4 | 4 | 4 | 4 | 3+ |
| Ningchun 4 | Sonora 64/Hongtu | 4 | 4 | 3− | 4 | 4 | 4 |
| Linmai 35 | Yong 2H15//Gui 86101/79531-1 | 4 | 4 | 4 | 4 | 4 | 1 |
| Xihan 2 | 8917C/Qinmai 3/72114 | 4 | 4 | 4 | 3 | 2 | ; |
| Dingxi 38 | RFMIII-101-A/Dingxi 32 | ;1 | 0 | 1 | 0 | 0 | 0 |
| Ganchun 21 | Aibai/Zhangchun 11//2014/82166-1- 2//Zhangchun 17 | 4 | 4 | 1 | 4 | 4 | ; |
| Dingxi 40 | 8152-8/Yong 257 | 4 | 3 | 4 | 1 | 4 | 4 |
| Wuchun 4 | 80-62- 3/7586//Rye//India Aisheng/Liaochun 10/Paulin | 0 | 0 | 0 | 1 | 0 | 0 |
| Wuchun 3 | Yi 5/Shi 857 | 4 | 4 | 4 | 3+ | 4 | 3 |
| Jinchun 5 | Shanqianhong/Funo | ; | ; | 0 | 2 | 0 | 1 |
| Gansu 26 | Unknown | 1+ | ;1 | 1 | 1 | 1 | 2 |
| Linmai 33 | 92 Yuan 11/Guinong 20 | 1 | 1 | ; | 0 | 1 | ; |

Xu et al. (2017), *PeerJ*, DOI 10.7717/peerj.4146

**Table 4** (*continued*)

| Cultivars (lines) | Pedigree | Infection types[a] | | | | | |
|---|---|---|---|---|---|---|---|
| | | 21C3CTHQM | 21C3CFHQC | 34MKGQM | 34MKGSM | 34C3RTGQM | 34C3MTGQM |
| Longchun 33 | Longchun 19/Longchun 23 | 4 | 1 | 3 | 1 | 4 | 0 |
| Jiuchun 6 | Jiu 96159/Jiu 9061 | ; | 0 | 0 | 0 | 1+ | 0 |
| Longchun 27 | 8858-2/Longchun 8 | 1 | 1+ | 1 | 1 | ;1 | 1 |
| Linmai 34 | 94 Xuan 4149/Guinong 20//82316/Linmai 26 | 0 | 0 | 0 | 0 | 2 | ; |
| Dingfeng 12 | Tal 73-3/Mota//Dingfeng 1 | 0 | 1+ | 2 | 2 | 1 | ;1 |
| Dingfeng 16 | 8447/CMS420 | 4 | 3 | 2 | 1 | 4 | ; |
| Zhangchun 21 | Gaoyuan 602/I 97-2//Gaoyuan 602 | 1 | 1 | 0 | ; | 1+ | 0 |
| Wuchun 6 | 80-62-3/Ningchun 4//Rye/India Aisheng/Liaochun 10//Paulin | 0 | 0 | ; | 2 | 0 | 1− |
| Lantian 23 | SXAF4-7/87-121 | 3+ | ; | 4 | 1 | 0 | ; |
| Lantian 19 | Mega/Lantian 10 | 4 | 4 | 4 | 4 | 4 | 4 |
| Lantian 25 | 95-173-4/Baofeng 6 | 3+ | 0 | 4 | 4 | 0 | 4 |
| Lantian 13 | A21//832809/872121-7 | ; | 4 | 3 | 4 | 0 | 4 |
| Xifeng 27 | 83183-1-3-1/CA837 | ; | 2 | 1 | 1+ | ; | 1+ |
| Lantian 26 | Flansers/Lantian 10 | 0 | 2 | 1 | 1+ | 1 | 1 |
| Longjian 101 | 85(1)F3 Xuan (2)-4/Shanhan 8968//85-173-12-2 | 4 | 1 | 4 | 4 | 4 | 4 |
| Hangxuan 1 | Unknown | 0 | 0 | 0 | 0 | 0 | 0 |
| Lantian 14 | Qingshang 895/Zhongliang 17 | 0 | 1+ | 0 | 0 | 0 | ; |
| Lantian 31 | Long Bow/Lantian 10 | 0 | 1 | 3− | 3 | 2 | 3 |
| Pingliang 42 | tal Changwu 131/Pingliang 38/82(51) | ;1 | 3− | 3− | 2 | 4 | 3 |
| Xifeng 20 | Xifeng 18/CA8055 | 1 | 3− | 2 | 2 | 1 | 1 |
| Longyu 4 | Xifeng 20/Zhong 210 | 0 | 1 | 2 | 2 | 1 | 1 |
| Changwu 131 | 7014-5/Zhongsu 68//F16-71 | 4 | 4 | 4 | 4 | 4 | 4 |
| Zhongliang 18 | Kangyin 655/Elytrigia trichophora//Jingai 21 | 4 | 3 | 0 | 1 | 4 | 4 |
| Zhongliang 22 | Zhong$_5$/S$_{394}$//Xiannong 4 | 0 | 0 | ;1 | 1 | 0 | 0 |
| Lantian 10 | Xifeng 16/Predgornajia/68286-0-1-1 | ; | 2 | 0 | 1 | 1 | 1 |
| Tianxuan 39 | Unknown | 1 | 1+ | ;1 | 0 | 0 | 1 |
| Huandong 6 | Unknown | 4 | 0 | 4 | 0 | 4 | 3 |
| Longjian 196 | 64035/Taiyuan 89/Qinnong 4 | 4 | 4 | 4 | 4 | 4 | 4 |
| Lantian 30 | 95-111-3/Shan167 | 1 | 2 | 2 | 3 | 1 | 2 |
| Longnan 2000-8-2-1 | Unknown | 0 | 1 | 0 | ;1 | 1 | 2 |
| Longjian 301 | DW803/7992 | 1+ | 1 | ; | 1+ | 1 | 2 |

Xu et al. (2017), *PeerJ*, DOI 10.7717/peerj.4146

**Table 4** (*continued*)

| Cultivars (lines) | Pedigree | Infection types[a] | | | | | |
|---|---|---|---|---|---|---|---|
| | | 21C3CTHQM | 21C3CFHQC | 34MKGQM | 34MKGSM | 34C3RTGQM | 34C3MTGQM |
| Longyu 2 | Longdong 3 //82(348)/9002-1-1 | 0 | 1 | 1 | 1 | 1 | 1 |
| Longjian P430 | Unknown | 0 | 1 | ; | 1 | 1 | 0 |
| Longjian 103 | Longjian 127/Mo(W)697 | 4 | 4 | 4 | 2 | 4 | 2 |
| Lantian 29 | 82F-37/83-44-20//8380 | 4 | 3 | 4 | 4 | 4 | 3 |
| Lan 092 | Unknown | 0 | 2 | 1− | 4 | 1 | 0 |
| Qingnong 1 | 7084/2037 | 4 | 4 | 4 | 3 | 4 | 3+ |
| Pingyuan 50 | Local cultivar | 3+ | 4 | 4 | 4 | 4 | 4 |
| Longyuan 034 | Unknown | 0 | 2 | 0 | 1+ | 0 | 1 |
| Lan 05-9-1-4 | Unknown | 4 | 4 | 4 | 2 | 4 | 3+ |
| Gandong 017 | Unknown | 0 | 2 | 2 | ;1 | 0 | 1 |
| Longjian 19 | Jinan 2/Qinnong 4 | 4 | 3 | 4 | 3 | 4 | 4 |
| Lantian 24 | 92R137/87-121-2 | 4 | 0 | 4 | 4 | 4 | 2 |
| 863-13 | Xiannong 4/Tianxuan 42 | 0 | 0 | 0 | 0 | 0 | 0 |
| 01-426e-1 | Unknown | 3+ | 4 | 3 | 4 | 4 | 3 |
| Tian 01-29 | Unknown | ; | 2 | 2 | 2 | ; | 2 |
| Tian 01-104 | Unknown | 4 | 4 | 4 | 3− | 4 | 4 |

**Notes.**

[a]Infection types (ITs): are based on a 0-to-4 scale where ITs of 0, ;, 1, and 2 are indicative of a resistant (low) response and ITs of 3 or 4 of a susceptible (high) response; Symbols + and − indicate slightly larger and smaller pustule sizes, respectively (*Stakman, Stewart & Loegering, 1962*).

Xu et al. (2017), *PeerJ*, DOI 10.7717/peerj.4146

**Table 5  Amplification results for the known *Sr* genes by markers.**

| Line | Sr gene | Source | Sr2 Xgwm533 | Sr2 csSr2 | Sr24 Sr24#12 | Sr24 Sr24#50 | Sr25 Gb | Sr26 Sr26#43 | Sr31 SCSS30.2$_{576}$ | Sr31 Iag95 | Sr38 VENTRIUP-LN2 | Sr38 URIC-LN2 |
|---|---|---|---|---|---|---|---|---|---|---|---|---|
| ISr5-Ra | 5 | 11Aberdeen | −[a] | − | − | − | − | − | − | − | − | − |
| CnS_T_mono_der | 21 | 11Aberdeen | − | − | − | − | − | − | − | − | − | − |
| Vernstine | 9e | 11Aberdeen | − | − | − | − | − | − | − | − | − | − |
| ISr7b-Ra | 7b | 11Aberdeen | − | − | − | − | − | − | − | − | − | − |
| IS11-Ra | 11 | 11GH | − | − | − | − | − | − | − | − | − | − |
| ISr-Ra | 6 | 11GH | − | − | − | − | − | − | − | − | − | − |
| ISr8a-Ra | 8a | 11Aberdeen | − | − | − | − | − | − | − | − | − | − |
| CnSr9g | 9g | 10Aberdeen | − | − | − | − | − | − | − | − | − | − |
| W2691SrTt-1 | 36 | 11GH | − | − | − | − | − | − | − | − | − | − |
| W2691Sr9b | 9b | 11Aberdeen | − | − | − | − | − | − | − | − | − | − |
| BtS30Wst | 30 | 11Aberdeen | − | − | − | − | − | − | − | − | − | − |
| Combination VII | 17+13 | 11Aberdeen | − | − | − | − | − | − | − | − | − | − |
| ISr9a-Ra | 9a | 11Aberdeen | − | − | − | − | − | − | − | − | − | − |
| ISr9d-Ra | 9d | 11Aberdeen | − | − | − | − | − | − | − | − | − | − |
| W2691Sr10 | 10 | 11Aberdeen | − | − | − | − | − | − | − | − | − | − |
| CnsSrTmp | Tmp | 11Aberdeen | − | − | − | − | − | − | − | − | − | − |
| LcSr24Ag | 24 | 11Aberdeen | − | − | + | + | − | − | − | − | − | − |
| Sr31/6*LMPG | 31 | 11Aberdeen | − | − | − | − | − | − | + | + | − | − |
| Trident | 38 | 11Aberdeen | − | − | − | − | − | − | − | − | + | + |
| McNair 701 | McN | Griffey 2010 | − | − | − | − | − | − | − | − | − | − |
| Line E | – | 09AB | − | − | − | − | − | − | − | − | − | − |
| Acme | 9g | 09AB | − | − | − | − | − | − | − | − | − | − |
| Siouxland | 24+31 | 2011 Baenzinger | − | − | + | + | − | − | + | + | − | − |
| Sisson | 31+36 | Griffey 2010 | − | − | − | − | − | − | + | + | − | − |
| SwSr22T.B. | 22 | 12GH | − | − | − | − | − | − | − | − | − | − |
| Agatha/9*LMPG | 25 | 08AB | − | − | − | − | + | − | − | − | − | − |
| Eagle | 26 | 10AB | − | − | − | − | − | + | − | − | − | − |
| 73,214,3-1/9*LMH? | 27 | 08AB | − | − | − | − | − | − | − | − | − | − |
| Federation*4/Kavl | 31 | 10AB | − | − | − | − | − | − | + | + | − | − |
| ER 5155 | 32 | 10AB | − | − | − | − | − | − | − | − | − | − |

Xu et al. (2017), *PeerJ*, DOI 10.7717/peerj.4146

**Table 5** (*continued*)

| Line | *Sr* gene | Source | *Sr2* Xgwm533 | *Sr2* csSr2 | *Sr24* Sr24#12 | *Sr24* Sr24#50 | *Sr25* Gb | *Sr26* Sr26#43 | *Sr31* SCSS30.2$_{576}$ | *Sr31* Iag95 | *Sr38* VENTRIUP-LN2 | *Sr38* URIC-LN2 |
|---|---|---|---|---|---|---|---|---|---|---|---|---|
| Tetra Canthatch/A? | *33* | 09AB | − | − | − | − | − | − | − | − | − | − |
| Mq(2)5XG2919 | *35* | 10AB | − | − | − | − | − | − | − | − | − | − |
| W3563 | *37* | 09Aberd | − | − | − | − | − | − | − | − | − | − |
| L6082 | *39* | 10AB | − | − | − | − | − | − | − | − | − | − |
| L6088 | *40* | 10AB | − | − | − | − | − | − | − | − | − | − |
| TAF 2 | *44* | 10AB | − | − | − | − | − | − | − | − | − | − |
| DAS15 | *47* | 10AB | − | − | − | − | − | − | − | − | − | − |
| Satu | *Satu* | 09Aberd | − | − | − | − | − | − | − | − | − | − |
| TAM 107-1 | *1A.1R* | 12GH | − | − | − | − | − | − | − | − | − | − |
| Fed*3/Gabo*21BI | *R* | 10AB | − | − | − | − | − | − | − | − | − | − |
| Iumillo | *9g,12,+* | 09GH | − | − | − | − | − | − | − | − | − | − |
| Leeds | *9e,13,+* | | − | − | − | − | − | − | − | − | − | − |
| Hope | *2* | | + | − | − | − | − | − | − | − | − | − |
| ST464 | *13* | 08GH | − | − | − | − | − | − | − | − | − | − |
| Q21861 | *Rpg1,4,5* | 04NewZealand | − | − | − | − | − | − | − | − | − | − |

**Notes.**

[a]Symbol '+' indicates the cultivar (line) carry the tested genes; '−' indicates that the cultivar (line) does not carry the tested genes.

# PeerJ

**Table 6 Molecular detection of resistance genes *Sr2*, *Sr24*, *Sr25*, *Sr26*, *Sr31*, and *Sr38* in the 75 wheat cultivars (lines).**

| Cultivars (lines) | Sr2 | Sr2 | Sr24 | Sr24 | Sr25 | Sr26 | Sr31 | Sr31 | Sr38 | Sr38 |
|---|---|---|---|---|---|---|---|---|---|---|
| | *Xgwm533* | *csSr2* | *Sr24#12* | *Sr24#50* | *Gb* | *Sr26#43* | *SCSS30.2$_{576}$* | *Iag95* | *URIC-LN2* | *VENTRIUP-LN2* |
| Ningchun 39 | −[a] | − | − | − | − | − | − | − | − | − |
| Dingfeng 10 | − | − | − | − | − | − | − | − | − | − |
| Linmai 32 | − | − | − | − | − | − | − | − | − | − |
| Wuchun 8 | + | − | − | − | − | − | − | − | − | − |
| Wuchun 7 | − | − | − | − | − | − | − | − | − | − |
| Dingxi 41 | − | − | − | − | − | − | + | + | − | − |
| Longchun 31 | − | − | − | − | − | − | + | + | − | − |
| Longchun 22 | − | − | − | − | − | − | − | − | − | − |
| Ganchun 25 | − | − | − | − | − | − | + | + | − | − |
| Longchun 25 | − | − | − | − | − | − | + | + | − | − |
| Longchun 23 | − | − | − | − | − | − | + | + | − | − |
| Longchun 26 | + | − | − | − | − | − | + | + | − | − |
| Ganchun 24 | + | − | − | − | − | − | + | + | − | − |
| Yinchun 9 | + | − | − | − | − | − | + | + | − | − |
| Longchun 28 | − | − | − | − | − | − | − | − | − | − |
| Wuchun 5 | − | − | − | − | − | − | − | − | − | − |
| Ganchun 20 | − | − | − | − | − | − | − | − | − | − |
| Ningchun 4 | − | − | − | − | − | − | − | − | − | − |
| Linmai 35 | − | − | − | − | − | − | − | − | − | − |
| Xihan 2 | − | − | − | − | − | − | − | − | − | − |
| Dingxi 38 | − | − | − | − | − | − | − | − | + | + |
| Ganchun 21 | + | − | − | − | − | − | − | − | − | − |
| Dingxi 40 | − | − | − | − | − | − | − | − | − | − |
| Wuchun 4 | − | − | − | − | − | − | − | − | − | − |
| Wuchun 3 | − | − | − | − | − | − | − | − | − | − |
| Jinchun 5 | − | − | − | − | − | − | + | + | + | + |
| Gansu 26 | − | − | − | − | − | − | + | + | + | + |
| Linmai 33 | − | − | − | − | − | − | − | − | + | + |
| Longchun 33 | + | − | − | − | − | − | − | − | − | − |
| Jiuchun 6 | + | − | − | − | − | − | − | − | + | + |
| Longchun 27 | + | − | − | − | − | − | + | + | − | − |
| Linmai 34 | − | − | − | − | − | − | − | − | − | − |
| Dingfeng 12 | + | − | − | − | − | − | − | − | − | − |
| Dingfeng 16 | − | − | − | − | − | − | − | − | − | − |
| Zhangchun 21 | − | − | − | − | − | − | + | + | − | − |
| Wuchun 6 | + | − | − | − | − | − | − | − | − | − |
| Lantian 23 | − | − | − | − | − | − | − | − | − | − |
| Lantian 19 | − | − | − | − | − | − | − | − | − | − |
| Lantian 25 | − | − | − | − | − | − | − | − | − | − |

*(continued on next page)*

**Table 6** (*continued*)

| Cultivars (lines) | Sr2 | Sr2 | Sr24 | Sr24 | Sr25 | Sr26 | Sr31 | Sr31 | Sr38 | Sr38 |
|---|---|---|---|---|---|---|---|---|---|---|
| | Xgwm533 | csSr2 | Sr24#12 | Sr24#50 | Gb | Sr26#43 | SCSS30.2$_{576}$ | Iag95 | URIC-LN2 | VENTRIUP-LN2 |
| Lantian 13 | − | − | − | − | − | − | − | − | − | − |
| Xifeng 27 | − | − | − | − | − | − | + | + | − | − |
| Lantian 26 | − | − | − | − | − | − | + | + | − | − |
| Longjian 101 | − | − | − | − | − | − | − | − | − | − |
| Hangxuan 1 | − | − | − | − | − | − | − | − | + | + |
| Lantian 14 | + | − | − | − | − | − | + | + | + | + |
| Lantian 31 | − | − | − | − | − | − | − | − | − | − |
| Pingliang 42 | − | − | − | − | − | − | − | − | − | − |
| Xifeng 20 | − | − | − | − | − | − | − | − | − | − |
| Longyu 4 | − | − | − | − | − | − | − | − | − | − |
| Changwu 131 | − | − | − | − | − | − | − | − | − | − |
| Zhongliang 18 | + | − | − | − | − | − | − | − | − | − |
| Zhongliang 22 | − | − | − | − | − | − | + | + | + | + |
| Lantian 10 | − | − | − | − | − | − | + | + | + | + |
| Tianxuan 39 | − | − | − | − | − | − | + | + | − | − |
| Huandong 6 | − | − | − | − | − | − | − | − | − | − |
| Longjian 196 | − | − | − | − | − | − | − | − | − | − |
| Lantian 30 | − | − | − | − | − | − | − | − | − | − |
| Longnan-2000-8-2-1 | − | − | − | − | − | − | − | − | − | − |
| Longjian 301 | − | − | − | − | − | − | + | + | − | − |
| Longyu 2 | − | − | − | − | − | − | + | + | − | − |
| Longjian P430 | − | − | − | − | − | − | + | + | − | − |
| Longjian 103 | − | − | − | − | − | − | − | − | − | − |
| Lantian 29 | − | − | − | − | − | − | − | − | − | − |
| Lan 092 | − | − | − | − | − | − | − | − | − | − |
| Qingnong 1 | − | − | − | − | − | − | − | − | − | − |
| Pingyuan 50 | − | − | − | − | − | − | − | − | − | − |
| Longyuan 034 | − | − | − | − | − | − | + | + | − | − |
| Lan 05-9-1-4 | − | − | − | − | − | − | − | − | − | − |
| Gandong 017 | − | − | − | − | − | − | + | + | − | − |
| Longjian 19 | − | − | − | − | − | − | − | − | − | − |
| Lantian 24 | − | − | − | − | − | − | − | − | − | − |
| 863-13 | − | − | − | − | − | − | + | + | − | − |
| 01-426e-1 | + | − | − | − | − | − | − | − | − | − |
| Tian 01-29 | − | − | − | − | − | − | + | + | − | − |
| Tian 01-104 | − | − | − | − | − | − | − | − | − | − |

**Notes.**
[a]Symbol '+' indicates the cultivar (line) carry the tested genes; '−' indicates the cultivar (line) don't carry the tested genes.
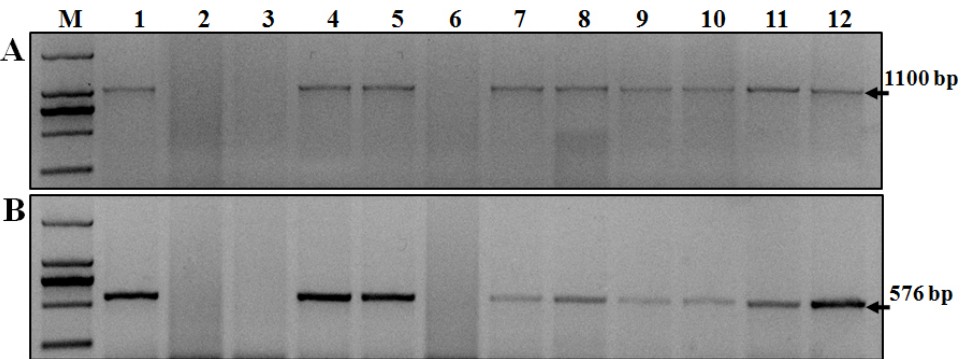

**Figure 2** **Amplification result for parts of wheat varieties with markers *SCSS30.2*$_{576}$ and *Iag95*.** (A) *Iag95*. (B) *SCSS30.2*$_{576}$. Lane 1–11, Monogenic *Sr31*, Little Club, Wuchun 7, Dingxi 41, Longchun 31, Longchun 22, Ganchun 25, Longchun 25, Longchun 23, Longchun 26, Ganchun 24, Yinchun 9, 'M' indicates 2,000 bp DNA ladder and black arrow indicates the position of the specific band.

### *Sr25* screening

Because of the resistance of *Sr25* to the new race Ug99 and related strains, a dominant marker *Gb* was developed for haplotyping *Sr25*, (*FAO, 2017*; *Liu et al., 2010*; *Pretorius et al., 2000*). The presence of the marker was confirmed by detection of a 130-bp fragment. The PCR results indicated that the 130-bp band was only amplified using the *Sr25*-positive line Agatha/9*LMPG (monogenic *Sr25*) genomic DNA (*Liu et al., 2010*; *Yu et al., 2010*), but not with other cultivar DNA samples, indicating that all 75 lines from Gansu Province examined lack *Sr25*.

### *Sr26* screening

Stem rust resistance gene *Sr26* was transferred into the long arm of wheat chromosome 6A from *Thinopyrum ponticum* (*Mago et al., 2005*). Although the cultivars carrying *Sr26* displayed resistance to all the dominant *Pgt* races in China, it is not utilized in wheat breeding. A dominant STS marker *Sr26#43* was developed for detecting this wheat stem rust resistance gene and a 207-bp band was amplified in wheat lines with *Sr26* (*Mago et al., 2005*). Marker *Sr26#43* was used to detect this fragment in tested wheat cultivars. No any visible band was detected, suggesting that these varieties do not carry *Sr26*, as expected.

### *Sr31* screening

Two markers, *SCSS30.2*$_{576}$ and *Iag95*, linked to resistance gene *Sr31* were used for detecting these locus. *SCSS30.2*$_{576}$ amplified a 576-bp fragment and marker *Iag95* amplified an 1,100-bp PCR fragment in *Sr31*-carrying lines such as *Sr31*/6*LMPG and Siouxland (Fig. 2). No fragment was amplified in the negative control LC. These two markers were used to detect *Sr31* in the tested cultivars. The result showed that these two fragments were detected in the 25 tested cultivars (Table 6).

### *Sr38* screening

The *Lr37*-*Sr38*-*Yr17* rust resistance gene cluster was transferred to the short arm of bread wheat chromosome 2AS from a segment of *Triticum ventricosum* (Tausch) Cess.

chromosome 2NS (*Helguera et al., 2003*). The 2NS-specific primer *VENTRIUP-LN2* and 2AS-specific primer *URIC-LN2* were developed to detect this rust resistance gene cluster in commercial wheat cultivars and 262-bp and 285-bp PCR products were amplified in wheat line carrying *Lr37-Sr38-Yr17*, whereas none of these amplification products were found in negative control LC (without *Lr37-Sr38-Yr17*). In this study, both 262-bp and 285-bp PCR fragments were amplified in nine wheat cultivars, suggesting that these wheat cultivars carried *Sr38* (Table 6).

## DISCUSSION

The broad-spectrum wheat stem rust resistance gene *Sr2* confers adult plant resistance to stem rust and is located on chromosome arm 3BS. It originated in tetraploid Yaroslav emmer (*T. dicoccum*) and later was transferred to the susceptible bread wheat 'Marquis' in the 1920s (*McFadden, 1930*). Several varieties with *Sr2* were cultivated worldwide (*Singh et al., 2011*). Markers *Xgwm533* and *csSr2* were used to detect *Sr2* in wheat cultivars from Gansu. However, marker *csSr2* failed to predict *Sr2*. Only marker *Xgwm533* amplified a 120-bp band in the positive control and 13 tested cultivars, but the 120-bp band also occurred in many North American and CIMMYT lines which are considered not to have *Sr2*. Therefore, it is difficult to conclude that all the accessions that showed a 120-bp fragment size for this marker carry *Sr2*.

The stem rust resistance gene *Sr24* is completely associated with leaf rust resistance gene *Lr24*. It has been widely used in wheat breeding programs worldwide, since it was introgressed into wheat lines (*McIntosh, Wellings & Park, 1995*). Gene *Sr24* was ineffective to some variants of Ug99 but is effective to the new races TKTTF, TTTTF, and many *Pgt* races in China (*Bhattacharya, 2017*; *Han, Cao & Sun, 2010*). Therefore, two markers, *Sr24# 12* and *Sr24#50*, developed by *Mago et al. (2005)* were used to detect the gene in Gansu wheat cultivars in this study. Surprisingly, no wheat cultivars carried this gene. However, it is reported that Chinese wheat cultivars in other provinces carry *Sr24* (*Cao et al., 2007*; *Li et al., 2016b*).

Wheat plants carrying stem rust resistance gene *Sr25* were susceptible to several strains of Chinese *Pgt* races (*Cao et al., 2007*). *Sr25* and its linked leaf rust resistance gene *Lr19*, were transferred into wheat from *Thinopyrum ponticum* to wheat chromosomes 7D and 7A (*Friebe et al., 1994*; *Zhang et al., 2005*). The use of *Sr25-Lr19* was initially limited because of linkage with another *Th. ponticum* derived gene producing undesirably yellow flour. It has been further backcrossed into the Australian and CIMMYT wheat backgrounds with the mutant line (which contains *Sr25-Lr19*), but with white flour (*Bariana et al., 2007*; *Knott, 1980*). The use of this gene in wheat programs is increasing for its resistance to new races TTTTF and Ug99 race group, having potential yield increases under irrigated conditions (*FAO, 2017*; *Liu et al., 2010*; *Monneveux et al., 2003*; *Singh et al., 1998*). In this study, 75 wheat varieties from Gansu Province were examined for presence of marker *Gb*. The result showed that all 75 wheat varieties lack *Sr25*.

In Australia, *Sr26* has been released in the cultivar Eagle since 1971 (*Martin, 1971*). Later, other major cultivars including Flinders, Harrier, Kite, Takari, and Sunelg, were cultivated.

Lines containing the *Sr26* fragment are resistant to new stem rust pathogen races such as Ug99 and its associated strains. None of the cultivars had *Sr26* in the present study, as expected, and similar results were observed in our previous study (*Li et al., 2016a*).

The stem rust resistance gene *Sr31* on 1BL/1RS was transferred into the bread wheat from 'Petkus' rye (*Graybosch, 2001*). Since then a higher number of wheat cultivars carrying *Sr31* have been released in global wheat breeding (*Das et al., 2006*). It is reported that more than 60% ($1.3 \times 107$ hm$^2$) of the total wheat planting areas carried this translocation in China (*Jiang et al., 2007*). Although the gene is ineffective to Ug99 and related variants, it is also an effective gene against all *Pgt* races in China and the new races TKTTF and TTTTF. Molecular marker detection showed that 25 wheat cultivars carried *Sr31*. All these cultivars (lines) produced resistance ITs (0, ;, ;1, 1+, and 2) to all tested *Pgt* races, as expected. Moreover, pedigree tracking indicated that resistant materials carrying the 1BL/1RS translocation such as 'Kavkaz' and 'Rye' were widely used in wheat breeding in Gansu Province (*Cao et al., 2011*), revealing the origin of *Sr31* in these wheat varieties.

Rust resistance gene cluster *Yr17-Lr37-Sr38* was initially transferred into the winter bread wheat line 'VPM1' from *T. ventricosum* and was located in a 2NS/2AS translocation (*Bariana & McIntosh, 1993*; *Cao et al., 2007*; *Maia, 1967*). PCR assays using restriction fragment length marker *cMWG682* were developed for selecting the 2NS/2AS translocation in wheat cultivars (*Helguera et al., 2003*). *Sr38* became susceptible to new races related to Ug99 but no virulent *Pgt* race to *Sr38* has been found in China. The results showed that nine wheat cultivars carried the gene cluster. The resistance of these cultivars against the tested *Pgt* races might be attributed to this gene.

## CONCLUSION

Breeding resistant cultivars is an economic and effective way to protect wheat from disease. The development of molecular technology facilitated the identification and utilization of molecular markers for durable resistance breeding, leading to increased crop production. The molecular markers associated with *Sr2*, *Sr24*, *Sr25*, *Sr26*, *Sr31*, and *Sr38* were used to detect the occurrence of these genes in 75 major wheat cultivars (lines) in Gansu Province in this study. The results showed that 35 tested cultivars might carry one of these genes. This information can be used in breeding for stem rust resistance in the future.

## ACKNOWLEDGEMENTS

We appreciate very much Dr. Fangping Yang at Wheat Research Institute, Gansu Academy of Agricultural Sciences for providing the wheat cultivars.

### Funding

This study was supported by National Natural Science Foundation of China (No. 31701738); the Science and Technology Research Project of Education Department of Liaoning (No. LSNYB201614), the National Key Basic Research Program of China (No.

2013CB127701). The funders had no role in study design, data collection and analysis, decision to publish, or preparation of the manuscript.

## Grant Disclosures

The following grant information was disclosed by the authors:

National Natural Science Foundation of China: 31701738.

Science and Technology Research Project of Education Department of Liaoning: LSNYB201614.

National Key Basic Research Program of China: 2013CB127701.

## Competing Interests

The authors declare there are no competing interests.

## Author Contributions

- Xiao Feng Xu, Dan Dan Li, Yang Liu, Yue Gao, Zi Yuan Wang and Yu Chen Ma performed the experiments.
- Shuo Yang analyzed the data.
- Yuan Yin Cao conceived and designed the experiments, contributed reagents/materials/analysis tools.
- Yuan Hu Xuan conceived and designed the experiments, wrote the paper, prepared figures and/or tables, reviewed drafts of the paper.
- Tian Ya Li conceived and designed the experiments, performed the experiments, analyzed the data, contributed reagents/materials/analysis tools, wrote the paper, prepared figures and/or tables, reviewed drafts of the paper.

## Data Availability

The raw data is contained in Table 4 and the Supplemental Files.

## Supplemental Information

Supplemental information for this article can be found online at http://dx.doi.org/10.7717/peerj.4146#supplemental-information.

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
