# Peer review of "Evaluation and identification of stem rust resistance genes Sr2, Sr24, Sr25, Sr26, Sr31 and Sr38 in wheat lines from Gansu Province in China"

_PeerJ, doi:10.7717/peerj.4146_

## Round 0.1 · original submission · Major Revisions

· Academic Editor

Major Revisions

Please revise your manuscript according to the suggestions of all 3 reviewers. Please, note 2 pdf files with annotations to the manuscript have been supplied bu the reviewers, which should also be taken into account. Please, explain all changes in the revised manuscript in a letter to the editor.

·

Basic reporting

* There are some sentences that are ambiguous and must be corrected and written in clear English and must conform to professional standards of expression.

Examples: lines 114-116, 117-120, 131, 141-145, 173-174, 186-187

* Citation in the text has o follow the standards

Examples: line 182, 186-187

* Headers of some of the tables did not explained well the contents of the respective tables.

Examples: Tables 4 and 5 need footnote what is "+" and "-".
There is a correction for the headers of Tables 1, 2 and 3 that
is highlighted in the attached annotated PDF.
* Needs specific conclusion based on on the obtained result rather than just putting a general conclusion at the end of the abstract and at the last part of the conclusion section (highlighted in the annotated PDF).

Experimental design

* The materials and methods is good except line 69 needs some clarification.

Validity of the findings

* Needs a specific conclusion based on the obtained result and the original research question.

Additional comments

* Use the right scientific names
Example: line 30

* Better to keep the list of cultivars in he table rather than long listing (42 cultivars) in the text
Example: lines 104-109, 152-155

* Put marker names in italics
Example: lines 174-176, in some of the references

* There are some corrections and comments in the attached annotated PDF

Reviewer 2 ·

Basic reporting

The manuscript will profit if corrected by a native speaker.

References in the text and at the end are in agreement, as well as figures and tables

Experimental design

The manuscript is withing scope and aims of PeerJ, objectives of work are clear.

Material and method: lack of information about symptoms and scores which specified for each level of diseas. Even in figure 1 not all type of scores have been included.

More details about used markers is missing.

Missing data about used wheat cultivars and their pedigree, how they are related.

Validity of the findings

For this reviewer the amount of information provided is not sufficient for publishing in PeerJ, but if authors can provide more data about used cultivars, infection types using more races and more Sr markers, then the manuscript could be reconsidered.

Additional comments

The paper "Stem rust seedling resistance genes evaluation and identification of Sr2, Sr24, Sr26, Sr31 and Sr38 in wheat lines from Gansu Province in China" describes analysis of seedling resistance to 3 stem rust isolates at 75 local wheat varieties followed with marker analysis being diagnostic to 5 known stem rust resistance genes. Authors found that 42 or 66.0% of tested wheat varieties are resistant to tested races of Puccinia graminis f. sp. tritici, while molecular marker analysis showed that 13 cultivars likely carrying Sr2; 25 carrying Sr31; and 9 Sr38. No cultivar was found to have Sr26 and Sr24.

Reviewer 3 ·

Basic reporting

The English is not professional, but the attached revision will help. Raw data of seedling infection types was not shared.

Experimental design

Experimental design is sufficient.

Validity of the findings

Data largely robust, but seedling infection type data should be provided.

Additional comments

The manuscript is largely well done. Please address the specific comments in the attached revision to help improve the manuscript.

Annotated reviews are not available for download in order to protect the identity of reviewers who chose to remain anonymous.

---

## Round 0.2 · Minor Revisions

· Academic Editor

Minor Revisions

Your manuscript is close to acceptance. There are still some language issues. Please see comments of the reviewers.

Also correct:

Line 43:...planted...

Line 184: rephrase:...which cultivates white wheat flour.

Tables 5+6: Add to which Sr genes the respective markers are linked.

·

Basic reporting

Greatly amended.

Experimental design

Good.

Validity of the findings

Fair.

Additional comments

Line 288 to 291 need revision (words are repeated).

The molecular genetic markers linked to associated with Sr2, Sr24, Sr25, Sr26, Sr31, and Sr38 were used to detect the occurrence of these genes in 75 major wheat cultivars (lines) in Gansu Province in this study.

Change to:

"The molecular markers associated with Sr2, Sr24, Sr25, Sr26, Sr31, and Sr38 were used to detect the occurrence of these genes in 75 major wheat cultivars (lines) in Gansu Province in this study"

In general the conclusion seems need paraphrasing;

Suggestion:
Breeding resistant cultivars is an economic and effective way to protect wheat from disease. The development of molecular technology facilitated the identification and utilization of molecular markers for durable resistance breeding, leading to increased crop production. The molecular markers associated with Sr2, Sr24, Sr25, Sr26, Sr31, and Sr38 were used to detect the occurrence of these genes in 75 major wheat cultivars (lines) in Gansu Province in this study. The results showed that 35 tested cultivars might carry one of these genes. This information can be used in breeding for stem rust resistance in the future.

Reviewer 3 ·

Basic reporting

The manuscript meets PeerJ standards.

Experimental design

The manuscript meets PeerJ standards.

Validity of the findings

Data are robust. Conclusions agree with data.

Additional comments

The authors addressed all of my concerns. I have some additional edits to improve the English of the newly added sections of the manuscript:

Abstract – change “a donor parent” to “donor parents”

Line 125 – use “genes” instead of “gene”

Line 311 – use “adult plant resistance”

Line 312 – use “chromosome arm”

Line 350 – use “Wheat plants carrying stem rust resistance gene Sr25 were susceptible to several…”

Line 352-353 – change to “to wheat chromosomes 7D and 7A from Thinopyrum ponticum (references).”

Line 357-358 – use “, but with white flour (Bariana…”

Line 358-359 – “The use of this gene in wheat programs is increasing for… and the Ug99…. potential yield increases….conditions”

Line 416 – delete “linked to”

---

## Round 0.3 · accepted · Accept

· Academic Editor

Accept

Your manuscript can be accepted. There are a few language errors which should be corrected while in production:

line 30:.. world
line 185: ....in wheat programs is.....
line 180....transferred into wheat from Thinopyrum ponticum to wheat chromosomes 7D and 7A (citations).